# Renal Cell Carcinoma with or without Tumor Thrombus Invading the Liver, Pancreas and Duodenum

**DOI:** 10.3390/cancers13071695

**Published:** 2021-04-03

**Authors:** Javier González, Jeffrey J. Gaynor, Gaetano Ciancio

**Affiliations:** 1Department of Urology, Hospital General Universitario Gregorio Marañón, 28007 Madrid, Spain; fcojavier.gonzalez@salud.madrid.org; 2Department of Surgery, Miami Transplant Institute, University of Miami Miller school of Medicine, Miami, FL 33136, USA; jgaynor@med.miami.edu; 3Department of Surgery and Urology, University of Miami Miller School of Medicine, Jackson Memorial Hospital, Miami, FL 33136, USA

**Keywords:** renal cell carcinoma, tumor thrombus, metastasectomy, postoperative complications, oncological outcomes

## Abstract

**Simple Summary:**

Renal cell carcinoma rarely invades the surrounding visceral structures. While surgical extirpation has been the mainstay of treatment for the localized disease, the role of surgery in cases of venous involvement, adjacent invasion or distant metastasis remains controversial. Furthermore, the surgical option may represent a challenge. A large series of locally advanced renal cancer with involvement of the liver, pancreas, and/or duodenum, sometimes in conjunction with tumor thrombus extending inside the inferior vena cava is herein reported. Our series establishes the technical feasibility of this complex surgical procedure with acceptable complication rates, no perioperative death, and potential for durable response. With the use of new systemic therapy schedules, these patients will probably have a better opportunity of survival extension.

**Abstract:**

Background: The purpose of this study is to report the outcomes of a series of patients with locally advanced renal cell carcinoma (RCC) who underwent radical nephrectomy, tumor thrombectomy, and visceral resection. Patients and methods: 18 consecutive patients who underwent surgical treatment in the period 2003-2019 were included. Neoplastic extension was found extending into the pancreas, duodenum, and liver in 9(50%), 2(11.1%), and 7(38.8%) patients, respectively. Seven patients (38.8%) presented also inferior vena cava tumor thrombus level I (*n* = 3), II (*n* = 2), or III (*n* = 2). The resection was tailored according to the degree of invasiveness. Demographics, clinical presentation, disease characteristics, surgical details, 30-day postoperative complications, and overall survival (OS) were analyzed. Results: Median age was 56 years (range: 40–76). Median tumor size was 14.5 cm (range, 8.8–22), and 10 cm (range: 4–15) for those cases with pancreatico-duodenal and liver involvement, respectively. Median estimated blood loss (EBL) was 475 mL (range: 100–4000) and resulted higher for those cases requiring thrombectomy (300 mL vs. 750 mL). Nine patients (50%) required transfusions with a median requirement of 4 units (range: 2–8). No perioperative deaths were registered in the first 30 days. Overall complication rate was 44.4%. Major complications were detected in 6/18 patients (33.3%). Overall median follow-up was 24 months (range: 0–108). Five-year OS (actuarial) rate was 89.9% and 75%, for 9/11 patients with pancreatico-duodenal involvement and 6/7 patients with liver invasion, respectively. Conclusion: Our series establishes the technical feasibility of this procedure with acceptable complication rates, no deaths, and potential for durable response.

## 1. Introduction

Renal cell carcinoma (RCC) is the most common malignant tumor of the kidney. Over 73,750 new cases will be diagnosed in 2020 [1,2,3]. RCC has a myriad of presentations, infrequently extends into the inferior vena cava (IVC) [4,5] and approximately one third of RCC are diagnosed in advanced stages of the disease [6]. Most of the 14,830 estimated deaths from RCC for 2020, will be associated with these stages [1]. While surgical extirpation has been the mainstay of treatment for the localized disease [3], the role of surgery in cases of venous involvement, adjacent invasion or distant metastasis remains controversial and the surgical option may represent a challenge [4,5,7,8,9].

There are to date no clear guidelines for the surgical management of large renal masses with contiguous extension and/or synchronous metastases to surrounding organs such as the pancreas, duodenum or liver at the time of initial nephrectomy due in part to still insufficient data and the lack of prospective randomized trials. However, resection for metastatic RCC disease has increasingly been performed with acceptable morbidity and mortality rates in the last few years [8,9]. The use of surgical strategy as a first line of treatment in this context would serve a multiple purpose: (i) it provides a psychological benefit to the patient who may feel “treated”, (ii) it provides symptomatic relief in the event of the presence of symptoms (rather frequent in this subgroup) (iii) it provides enough tissue sample to be used for assessment on the heterogeneity of the disease (usually present) and on the morphological, immunohistochemical and molecular variants involved, serving commonly as a guide for subsequent systemic treatment (type of agent/s and treatment sequence among the many currently available), (iv) reduces the burden of disease, thus avoiding the immunological sink generated by the overwhelming amount of malignancy and enabling an effective immune response in the patient against it, and (v) such an aggressive resection has revealed a modest survival benefit in favor of this approach in opposition to observation or systemic treatment alone [10].

We present a contemporary series of 18 patients with locally advanced renal masses invading adjacent visceral structures in which complete removal of gross disease was achieved by means of radical nephrectomy in conjunction with thrombectomy, resection of pancreas, duodenum and/or liver with low morbidity and mortality rates by using the surgical techniques derived from the field of transplantation applied to these complex cases. With this approach in the front line, we provide for symptomatic relief in most our patients, obtain enough sample of tissue to guide further systemic treatment, and observe durable responses.

## 2. Materials and Methods

After obtaining Institutional Review Board approval and following the ethical principles of the Helsinki Declaration (as revised in 2013), a retrospective chart review was performed on 18 consecutive patients who underwent surgical treatment for large, aggressive, locally advanced RCC cases between June 2003 and November 2019 at our institution. All relevant data on demographics, clinical presentation, disease characteristics, surgical details, and postoperative complications were collected and analyzed. Postoperative complications were assessed using the Clavien-Dindo Classification System [11], and defined as occurring within a 30-days period of the intervention date. Overall survival (OS) was ascertained by the review of medical records or using the Social Security Death Index (SSDI) database [12] when necessary. Abdominal computed tomography and/or magnetic resonance imaging were used to diagnose the renal tumor, delineate the tumor thrombus inside the inferior vena cava, and depict the extent of the invasion to adjacent organs (Figure 1).

### Surgical Technique

Our transplant techniques for piggy-back liver mobilization and en-bloc mobilization of spleen and pancreas in order to facilitate the resection of large renal tumors have been previously described [4,5,7,13]. Briefly, the renal artery was ligated early during the surgery by medially mobilizing the involved kidney [14]. The goal of early renal artery ligation was to decompress collateral circulation and decrease blood loss during the procedure. The use of the maneuvers derived from transplantation surgery for either side, allowed wide exposure of the retroperitoneal space thus facilitating the removal of the renal mass and the resection of variable segments of other organs involved. All visceral resections were performed by a single surgeon and with curative intention, and tailored according to the particular degree of tumor invasiveness. Intracaval involvement required IVC exploration, thrombus withdrawal, and IVC resection/reconstruction as previously described [13,14,15].

At the time of resection, neoplastic extension was found extending outside the perirenal tissue into the pancreas in 9/18 patients (50%), mostly coming from the left side. The second portion of duodenum was affected in two right-sided tumors (18%), a left side tumor invaded the third and four portions of the duodenum in one patient (9%), and the remaining case (9%) showed tumor invasion extended into both pancreas and duodenum.

These 9 patients underwent variable pancreatic resections that included pancreatico-duodenectomy (i.e., Whipple procedure) (9%), partial or subtotal pancreatectomy (63%), and combined body and tail pancreatectomy (9%). One patient underwent spleen preservation along with distal pancreatectomy. However, the spleen and the ipsilateral adrenal were also excised en-bloc in 7 (63%) and 9 (100%) cases, respectively. The duodenum was preserved whenever possible, although partial resection was considered necessary in cases of exclusive serosal involvement (27%). Intracaval extension was additionally detected in 3 of the patients (27%); one case of level II tumor thrombus(TT) according to Neves and Zincke [16] (showing concomitant invasion of the left colon, thus requiring additional left hemicolectomy), and two cases of level III TT (i.e., IIIa and IIIb tumor thrombi according to our classification) [17].

Partially occluding (level I or II) thrombi were managed by tangential cavectomy and primary closure. Completely occluding (level II and III) tumor thrombi required circumferential cavectomy (i.e., segment located between the caval bifurcation and the inferior border of the major hepatic veins entrance) without vascular reconstruction. The level IIIb tumor thrombus was “milked” and controlled below the major hepatic veins before the cavectomy was attempted (Figure 2) [18].

RCC was infiltrating the right lobe in three patients. Partial right lobectomy was performed en-bloc with the renal tumor and adrenal gland in three patients, resection of segments two and three in one patient, and wedge-resection of the liver mass in other three patients. Four of these patients harbored also an IVC tumor thrombus (level I in *n* = 3, and level II in *n* = 1) that required additional thrombectomy, tangential cavectomy, and primary closure. Of note, one of the patients that underwent a partial right lobectomy had two previous unsuccessful ipsilateral renal artery embolization attempts complicated by massive bleeding before referred to our institution (Figure 3).

Extended lymph node dissection (including all the potentially affected palpable lymph nodes, and paracaval, inter aortocaval and paraaortic templates depending on the particular features of the case) completed the procedure in all cases.

## 3. Results

### 3.1. Patient Demographics

Over a period of 16 years, a total of 18 patients underwent radical nephrectomy, tumor thrombectomy (when present), and resection of visceral structures invaded by disease. Table 1 lists pertinent demographic information. The median age at the time of resection was 56 years (range: 40–76 years). There were seven males (38.9%) and 11 females (61.1%). Except for four cases (22%) in which an incidental mass was found on imaging, all the remaining cases were symptomatic (78%). Complaints in symptomatic patients ranged from abdominal pain (44%) to gastrointestinal (GI) bleeding (5.5%) due to tumor erosion into duodenum. One patient presented with collapsed lung due to endobronchial RCC metastases and required left upper lobe resection prior to the abdominal intervention [15]. None of the patients had obstructive jaundice. Median tumor size was 14.5 cm (range, 8.8–22 cm), and 10 cm (range: 4–15 cm) for those cases with pancreatico-duodenal and liver involvement, respectively.

### 3.2. Intra-Operative Outcomes

Overall median estimated blood loss (EBL) was 475 mL (range: 100–4000 mL). Median EBL for those requiring pancreatic (*n* = 11) and liver (*n* = 7) resections was 300 mL (range, 100–2500 mL) and 500 mL (range 200–4000), respectively; while median EBL for those having (*n* = 7) vs. not having (*n* = 11) IVC exploration (along with either liver resection, duodenal resection or pancreatectomy) was 750 mL (range, 200–4000 mL) and 300 mL (range 100–600), respectively. Nine of the 18 patients required blood transfusions during surgery or in the immediate postoperative period; the median transfusion requirement for the nine patients who required transfusions was four units of packed red blood cells (range: 2–8 units).

### 3.3. Surgical Outcomes

The intervention provided complete or partial symptomatic relief in 12/14 cases (85%) and 2/14 cases (15%), respectively. Fatigue improved considerably in all patients presenting it, but remained after the intervention in two of them (66%). No perioperative deaths were registered in this series during the first 30 days after the intervention. Overall complication rate was 44.4% (8/18 patients). Major complications (Clavien-Dindo grade ≥ III) were detected in 6 of the 18 patients (33.3%); 45% (5/11) of the patients with pancreatico-duodenal involvement and 14% (1/7) of the patients showing liver invasion. Postoperative complications included isolated intra-abdominal collection (2), wound infection (1), pancreatitis (1), pancreatic leak (1), and atrial fibrillation with pulmonary embolization that required prolonged intubation (1) for those patients with pancreatico-duodenal invasion, and postoperative intra-abdominal collection associated to pulmonary embolization that required prolonged intubation in one of the patients with liver involvement. Table 2 and Table 3 list operative outcomes and classified complications, respectively.

All tumors were classified as T4 according to the American Joint Committee on Cancer (AJCC) 2009 TNM staging system [19]. Table 4 lists pathological characteristics for all specimens. Among the specimens showing pancreatico-duodenal involvement, histological examination revealed no positive margins, and RCC conventional type in 9 (81%), chromophobe type in 1 (9%), and metastatic poorly differentiated carcinoma from the adrenal in 1 (9%). Specimens presenting liver involvement showed RCC conventional type in 5 patients (57%), chromophobe type in 1 (14%) and poorly differentiated squamous cell carcinoma in 1 (14%). Median tumor grade was higher (IV vs. III) for those tumors invading the pancreas or duodenum than for those presenting liver infiltration. However, neither categorization of Fuhrman grade nor identification of the source was possible in the case showing the squamous cell carcinoma invading the liver.

A total of 50 lymph nodes were removed from the 11 patients showing pancreatico-duodenal involvement. Only one of the patients showed invasion by RCC (conventional type). Twenty lymph nodes were removed from the seven patients showing liver involvement. Pathologic examination of these specimens resulted positive in two of these patients (RCC clear cell type and poorly differentiated squamous cell carcinoma, respectively).

### 3.4. Oncological Outcomes

Overall median follow-up was 24 months (range: 0–108 months). Two of the 11 patients with pancreatico-duodenal involvement returned to their local community physicians and were lost for further follow-up. Overall survival (OS) was ascertained in the remaining nine patients showing a five-year (actuarial) rate of 89.9%. Excluding the two patients with no post-surgical follow-up, the median survival time from the date of resection was 36 months (range: 13–108 months). One patient (9%) lived for nine years but developed multiple metastases during the follow-up (87 months) requiring additional resection and adjuvant systemic therapy (sunitinib); another patient (9%) remained alive after seven years but need systemic chemotherapy for the treatment of lymphoma at the time of last follow-up. The remaining six patients (66.7%) had no evidence of disease as of last visit.

One of the seven patients with liver involvement returned to his local community physician for follow-up. OS was ascertained in the remaining six patients showing a five-year (actuarial) OS rate of 75%. Excluding the single patient with no post-surgical follow-up, the median survival time from the date of resection was 24 months (range: 12–96 months). One patient died after two years of multiple metastases.

## 4. Discussion

The main purpose of this study was to review our experience in the removal of renal tumors in conjunction or not with tumor thrombus that required additional liver, pancreas and/or duodenum. We described the surgical outcomes of a contemporary series including a total of 18 consecutive patients. Eleven of these patients, underwent radical nephrectomy and resection of the pancreas and/or duodenum. Three of them presented a TT inside the IVC, making their surgical management even more complex. We also included seven patients with renal tumors invading the liver (four of them with TT involving the IVC) in whom simultaneous radical nephrectomy, removal of TT if necessary, and liver resection was performed. Our series of multiorgan resection was associated with a major complication rate of 33.3% (*n* = 6/18) in the absence of post-operative mortality. These outcomes confirm again the technical feasibility already established by other series [20,21]. More importantly, this study reconfirms that in experience hands and properly performed, this approach is worth the effort since shows curative intent and provides better quality of life by means of symptomatic relief.

In this regard, Yezhelyev et al. reported their experience of 25 patients that underwent simultaneous radical nephrectomy and major hepatectomy for RCC [20]. Eight of those patients presented a TT in the IVC. Direct liver invasion by contiguity was the most common indication for hepatectomy. Ten patients in these series (40%) presented postoperative complications including one death in the perioperative period. Karellas et al. also described their experience of 38 patients with pT4 RCC invading adjacent organs. The liver was the most commonly visceral structure resected en-bloc with RCC (*n* = 10), but they also reported pancreatic resection in six patients. They recorded two perioperative deaths, but no report of post-operative complications was provided. Only 1 patient was alive and free of disease at five years. The median time from surgical resection to death was 11.7 months. Their conclusion was that once advanced RCC involved the adjacent structures the prognosis was poor [21].

Eleven patients in our series showed either pancreatico-duodenal (*n* = 9) or isolated duodenal involvement (*n* = 2) at the time of the radical nephrectomy. Although the involvement of the pancreas from RCC is rare, is still more frequent that isolated duodenal involvement. In this regard, Margulis et al. reported their experience in the management of these cases [22], which resulted quite similar to the Karellas et al. [21] in terms of morbidity and perioperative mortality. They reported 12 out of 30 patients (40%) presenting pT4 disease at debut with direct invasion into adjacent organs demonstrated by preoperative imaging. The pancreas was involved in only three of these patients, none of them invading the duodenum. Despite the aggressive surgical approach, the disease recurred in 10 of those 12 patients (83.3%) at a median time of 2.3 months, and only five (41.6%) were alive at the time of the report [22].

Resection of RCC metastasizing into the duodenum has been rarely described in a number of case reports. Most of these metastases were metachronous and frequently coursed with upper GI bleeding, in the same way that one of the patients included in our series [23,24,25]. Conversely, duodenal resection at the time of the radical nephrectomy has been only reported once [26]. In our series, four of the patients had the duodenum resected at the time of the radical nephrectomy; including a Whipple procedure, two partial duodenal resections, and an excision of the third and fourth portions of the duodenum in conjunction with distal pancreatectomy and splenectomy.

RCC represents an entity with high potential for metastatic spreading. Recent epidemiological datasets report that approximately 17% of patients debut with metastases, and almost one third of the remaining patients will develop local recurrences or distant metastases during the follow-up [1]. Although metastasectomy in locally advanced or metastatic RCC has been used since the 1930s, to date no prospective randomized controlled trials have evaluated the clinical benefit of this approach. Conversely, surgical tumor debulking is currently supported by retrospective experiences showing rather favorable overall and progression free survival rates [27]. Actually, two recent systematic reviews suggested that complete metastasectomy is associated with better survival (range, 36.5–142 months vs. 8.4–27 months) and/or symptomatic control when compared to no or incomplete metastasectomy [28]. Comparable outcomes have been reported in the context of the resection of multiple metastatic sites, local/distant lymph nodes, and local recurrences [29]. In fact, disease-free intervals after complete resection may be notable, thus avoiding systemic treatment requirements at least in the first instance [30]. These observations are in the line of our experience with actuarial five-year OS rates reaching 89.9% and 75% for those patients showing pancreatico-duodenal and liver involvement, respectively.

The difference in terms of survival between these two subgroups seems notable in favor of the subgroup with exclusive pancreatico-duodenal involvement. Although in cases of pancreatico-duodenal involvement postoperative complications are more frequent, the actuarial five-year OS is approximately 15% higher. A possible explanation for these results would rely on differences in the pattern of disease dissemination between the subgroups. While pancreatic-duodenal involvement occurs mainly by direct contiguity, liver involvement may result from the coexistence of direct invasion or hematogenous dissemination. Probably, the latter would impact negatively the prognosis. Conversely, in cases in which pancreatic resection affected the cephalic region, surgical reconstruction after removal is technically more demanding and, with no doubt, more subject to complications than the resection of a portion of the liver parenchyma. Nevertheless, intraoperative bleeding in these cases is commonly more exuberant and difficult to manage intraoperatively.

Although there are no current clear guidelines regarding patient selection in order to determine who will benefit most from surgery, the decision-making regarding the resection of synchronous neighboring visceral metastases seems to rely on different factors such as comorbid patient conditions, performance status, a number of different prognostic risk factors, as well as number and location of the sites involved. According to the National Comprehensive Cancer Network guidelines, metastasectomy should be considered in those patients with clear or non-clear cell variants who initially present with primary RCC in conjunction with oligometastatic involvement or develop metastases after a prolonged free-disease interval from initial nephrectomy [31]. In addition, the European Society for Medical Oncology recommends that the decision-making process should be upon a multidisciplinary team trying to identify patients with adequate performance status, resectable oligometastatic disease, or low/intermediate risk in which complete resection is achievable [32], given that the best outcomes after metastasectomy have been reported in such cases. Only one of our patients showed a solitary lung metastasis that was resected before attempting the abdominal intervention.

All of them were considered suitable for surgery in terms of performance status and comorbidity, and the decision-making regarding the extent of the local resection was commonly made upon the direct vision of the operative field. Nevertheless, considering this possibility, extensive counseling was made in a particular basis upon the findings provided by preoperative cross-sectional imaging.

Doubts regarding complete resection predicting short-term failure for disease control has led to concepts combining surgery either with presurgical or adjuvant systemic therapy. The neoadjuvant approach followed by complete surgical excision has been evaluated in small retrospective series during the cytokine era [33]. The greater effect in terms of efficacy in response or downsizing the tumor burden (reaching a median reduction of 9-14% in primary tumor diameter) obtained with targeted therapies opened the gates for further multimodality treatment schedules in this context. Surgery following tyrosine kinase inhibitors is overall safe [34], making candidates initially unsuitable for complete resection amenable for surgical approach reconsideration if downsizing or significant response is confirmed after at least one cycle of targeted therapy [35,36]. Although not prospectively studied, acceptable outcomes have been reported with this approach in small retrospective series [34]. The largest experience is based on a 22-patient series from three different institutions in which primary excision associated with concomitant or deferred metastasectomy was performed after at least one cycle of tyrosine kinase inhibitors [37]. However, most of the patients included continued systemic treatment once recovered from surgery, making the evaluation of the long-term free-relapse survival periods observed difficult to attribute to either surgery or systemic treatment alone, or the combination of both treatment modalities acting synergically.

Currently, no role for systemic adjuvant therapy after metastasectomy has been established. A randomized, double-blinded, placebo-controlled multicenter phase-III trial conducted by the Eastern Cooperative Oncology Group-American College of Radiology Imaging Network group assessed the role of adjuvant pazopanib vs. placebo in patients with no evidence of disease after metastasectomy [38]. The patients received treatment for a year and were stratified according to the number of sites of resected disease as well as disease-free interval. The study was unable to meet a primary end-point for disease-free survival (HR 0.85; CI95% 0.55–1.31; *p* = 0.47), actually showing a trend to worse overall survival with pazopanib. Comparable results have been reported with sorafenib in the postmetastasectomy setting [39]. Currently, different ongoing trials are evaluating the use of sunitinib and a number of immune checkpoint inhibitors in this context, but there is not enough evidence to date to support their use over observation.

It remains unclear what role metastasectomy or surgical consolidation will play in the next future, given the morphing landscape of systemic immunotherapy combined with targeted therapy, and new data on different tumor molecular features acting as potential biomarkers for response to treatment. So far, the clinical experience should guide the recommendations for metastasectomy in these patients. Therefore, our surgical outcomes allow us to advocate in favor of a surgical approach if considered feasible and indicated. However, this study is not without limitations. The series herein reported is retrospective in nature and reflects the experience of a single high-volume surgeon from a referral center. Hence, these outcomes should be evaluated accordingly, given that may be not reproduced by teams of lower experience or working in different clinical settings.

## 5. Conclusions

We reported a large series of locally advanced RCC with involvement of the liver, pancreas, and/or duodenum, sometimes in conjunction with TT extending inside the IVC. Our series establishes the technical feasibility of this complex surgical procedure by means of the application of the techniques derived from transplant surgery with acceptable complication rates, no perioperative death, and potential for durable response. In addition, by using this strategy in the front line we obtain symptomatic relief in most of the patients and provide for an optimal tissue sample that may be used to guide the further systemic therapeutic approach required. With the use of new systemic therapy schedules, these patients will probably have a better opportunity of survival extension. In our opinion, the management of these rare cases should be safely entrusted to an experienced surgeon/surgical team, but exclusively in high volume referral centers, where complex procedures, such as the ones reported, can be successfully performed.

## Figures and Tables

**Figure 1 cancers-13-01695-f001:**
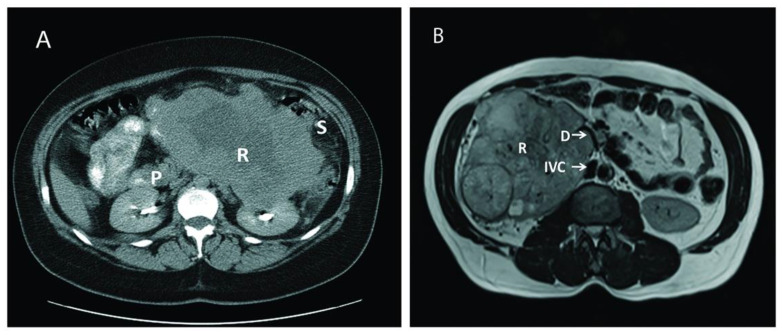
Computed tomography (CT) and magnetic resonance imaging (MRI) of two different patients with large renal mass. (**A**): CT of a large left renal mass with lack of soft tissue planes between pancreas, spleen, and splenic flexure; (**B**): MRI of a right large solid and cystic enhancing mass replacing the right renal parenchyma with mass effect and compression of proximal duodenum. P: Pancreas; R: Renal Tumor; S: Splenic flexure of colon; IVC: Inferior Vena Cava: D: Duodenum.

**Figure 2 cancers-13-01695-f002:**
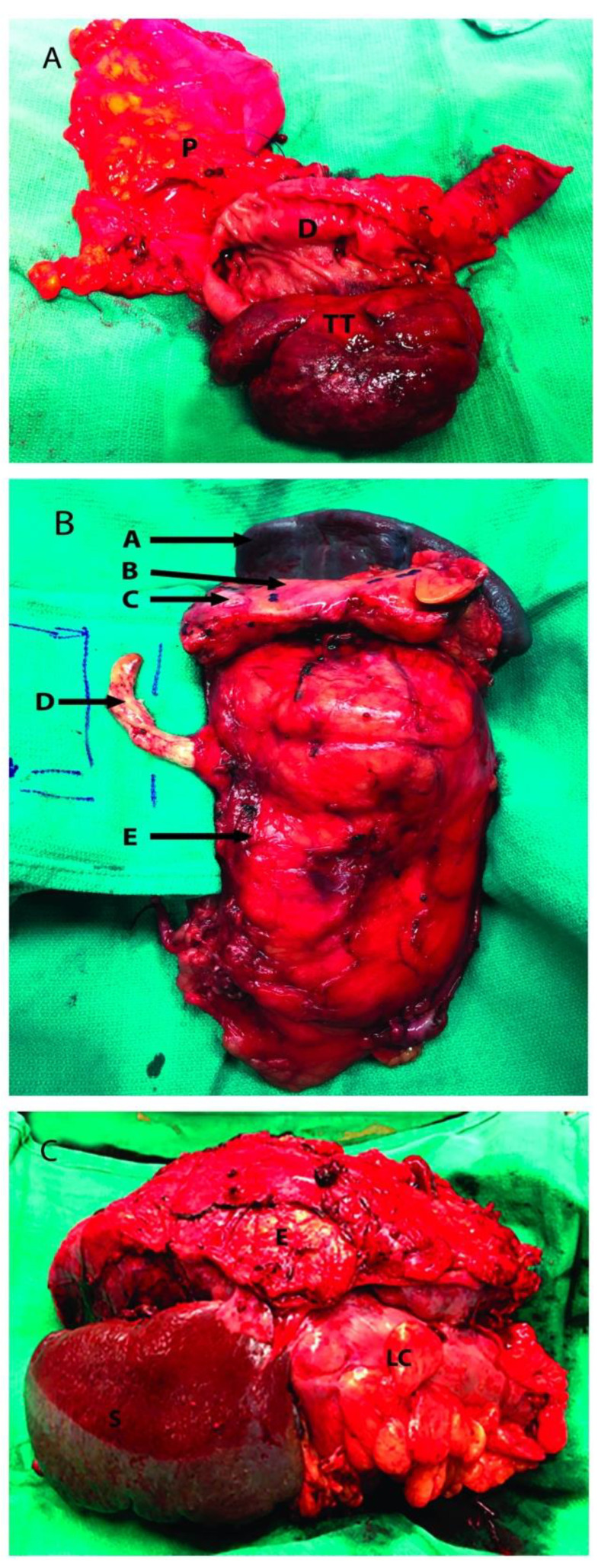
Renal tumor involving adjacent organs. (**A**): Head of the pancreas resected along with a portion of duodenum; (**B**): Left radical nephrectomy specimen with resection of IVC tumor thrombus, pancreas and spleen; (**C**): Left kidney tumor resected with adjacent organs, the distal pancreas was not visualized. P: Head of pancreas; D: 2nd portion of duodenum; TT: Tumor thrombus inside the second portion of the duodenum; A: Spleen; B: Pancreas; C: Pancreatic metastatic mass; D: Tumor Thrombus going inside the inferior vena cava; E: Kidney Tumor; S: Spleen; LC: Left Colon.

**Figure 3 cancers-13-01695-f003:**
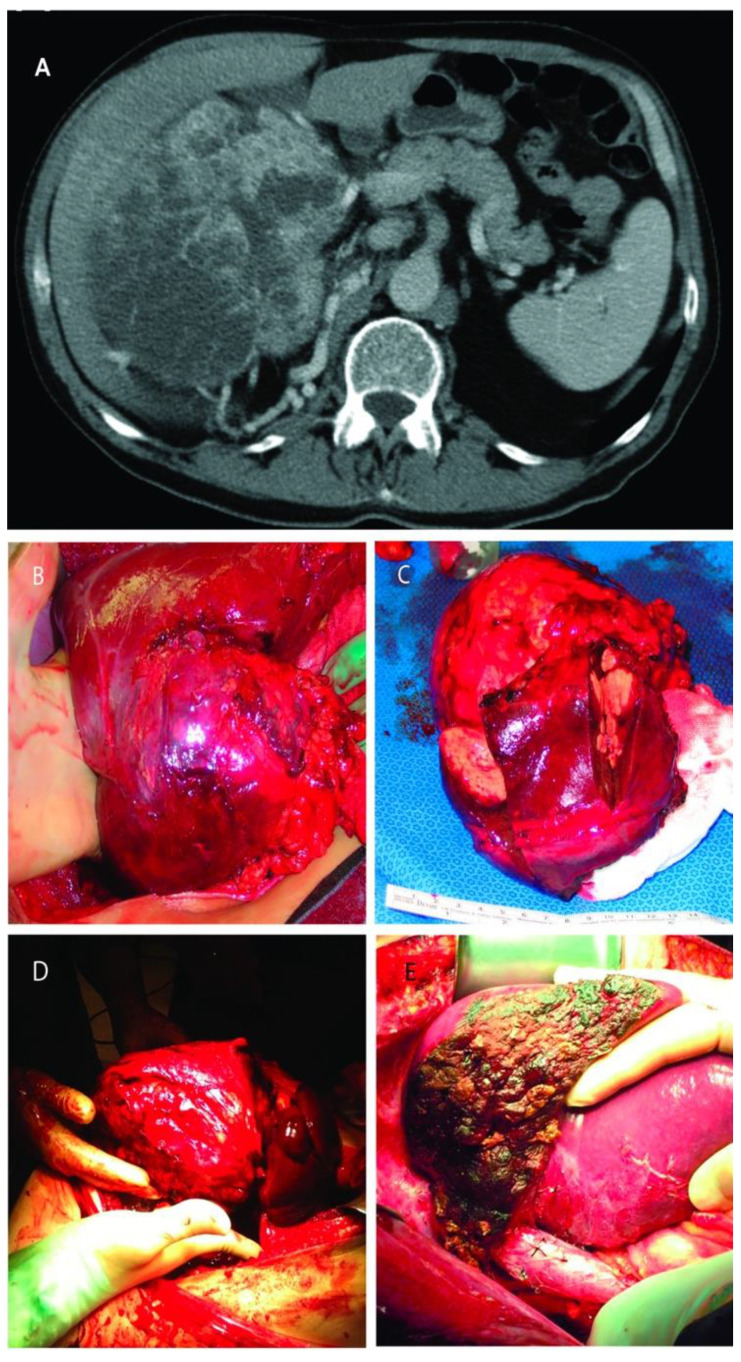
(**A**): Computed tomography of large right renal mass with collaterals from the liver; (**B**): Right renal mass with renal artery and vein ligated but still with blood flow from the liver; (**C**): The resected specimens of right renal mass with right lobe of the liver; (**D**): Right renal mass infiltrating the right lobe of the liver; (**E**): Remaining liver after almost complete right lobectomy.

**Table 1 cancers-13-01695-t001:** Demographic information.

Parameter	Value
No. of patients	18
Male, *n* (%)	7 (38.9%)
Female, *n* (%)	11 (61.1%)
Age, median (years)	56 (40–76)
Follow up, median (months)	24 months (0–108); 3 deaths at 13, 24, and 108 months post-surgery, respectively
Asymptomatic, *n* (%)	4 (22%)
Symptomatic, *n* (%)	14 (78%)
Abdominal pain	8 (42%)
Hematuria	3 (16.6%)
Fatigue	3 (16.6%)
GI bleeding	1 (5.5%)
Collapsed lung	1 (5.5%)

GI: gastrointestinal.

**Table 2 cancers-13-01695-t002:** Operative details and outcomes.

Parameter	Value
Side of lesion, *n* (%)	
Left	8 (38.8%)
Right	11 (61.2%)
IVC involvement ^1^, *n* (%)	
Level I	3 (16.6%)
Level II	2 (11.1%)
Level III ^2^	2 (11.1%)
Tumor size (cm)	
Pancreatico-duodenal involvement	14.5 (8.8–22)
Liver involvement	10 (4–15)
EBL, median (mL)	475
Pancreatico-duodenal involvement	300 (100–2500)
Liver involvement	500 (200–4000)
IVC exploration	750 (200–4000)
No IVC exploration	300 (100–600)
Transfusion, median (PRBC units)	1U for the total group of 18 patients; 4U among the 9 patients who required transfusions
Liver resection ^3^, *n* (%)	7 (38.8%)
Pancreaticoduodenectomy, *n* (%)	1 (5.5%)
Partial/subtotal pancreatectomy, *n* (%)	7 (38.8%)
Distal pancreatectomy, *n* (%)	1 (5.5%)
Other organs removed, *n* (%)	2 (11%)
Partial duodenum	3 (16.6%)
Spleen	7 (38.8%)
Adrenal gland	18 (100%)
Left colon	1 (5.5%)

IVC: inferior vena cava; PRBC: packed red blood cells; U: units; ^1^ Acoording to the Neves-Zincke Classification System; ^2^ One of the patients presented a level IIIa tumor thrombus (below the major hepatic veins), while the other presented a level IIIb tumor thrombus (at the level of the major hepatic veins); ^3^ Partial right lobectomy in 3 patients, resection of segments 2 and 3 in 1 patient, and wedge resection in 3 patients.

**Table 3 cancers-13-01695-t003:** Postoperative complications according to the Clavien-Dindo Classification System.

Complication	*n* (%)	Grade
Collection	3 (16.6%)	IIIa
Pulmonary ^1^	2 (11.1%)	Iva
Atrial fibrillation	1 (5.5%)	II
Wound infection	1 (5.5%)	II
Deep venous thrombosis	1 (5.5%)	IIIa
Pancreatitis	1 (5.5%)	II
Pancreatic leak	1 (5.5%)	IIIa

^1^ Pulmonary embolization with prolonged intubation.

**Table 4 cancers-13-01695-t004:** Pathological characteristics and survival outcomes.

Variable	Value
AJCC pT4, *n* (%)	18 (100%)
RCC conventional type, *n* (%)	14 (77.7%)
Fuhrman Grade, median	
Pancreatico-duodenal involvement	IV
Liver involvement	III
Lymph node metastases, *n* (%)	
Pacreatico-duodenal involvement	1 (9%)
Liver involvement	2 (28.5%)
Median survival from time of resection (months)	
Pancreatico-duodenal involvement	36 (13–108)
Liver involvement	24 (12–96)
Actuarial 5-yr OS (%)	
Pancreatico-duodenal involvement	84.6%
Liver involvement	75%

RCC: Renal cell carcinoma; yr: years; OS: overall survival.

## Data Availability

Data supporting reported results may be obtained upon request to the corresponding author.

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
