# Peer review of "Renal Cell Carcinoma with or without Tumor Thrombus Invading the Liver, Pancreas and Duodenum"

_cancers, 2021, doi:10.3390/cancers13071695_

Round 1

Reviewer 1 Report

The manuscript entitled ‘renal cell carcinoma with or without tumor thrombus invading the liver, pancreas and duodenum’ is interesting study and presented very well. This manuscript suitable for publications. However, minor revision required before publication. The comments to author are as follow

  1. Introduction has scope for further improvement to highlight the key advantage of the present study
  2. Conclusion should highlight the major outcome of study and how it will be contributing in the advancement of researcher area
  3. Figure 2 and 3 can be re-organized.
  4. Optional to author, it will be more easy to understand the goal of the study if author add a scheme or figure presenting overview of the present study at end of introduction.

Author Response

Reviewer #1

The manuscript entitled ‘renal cell carcinoma with or without tumor thrombus invading the liver, pancreas and duodenum’ is interesting study and presented very well. This manuscript suitable for publication. However, minor revision required before publication.

We thank the reviewer for taking the time to review our manuscript. We are pleased that this reviewer finds our experience useful and recommends its publication in this journal. We will try to answer his/her constructive comments in a point-by-point fashion as required.

The comments to author are as follows:

Introduction has scope for further improvement to highlight the key advantage of the present study

The aim of our study was to present our experience in the surgical management of a large volume series of patients harboring a rare entity. This experience is based on the application of a set of surgical maneuvers derived from transplant surgery to deal with these complex cases in a safely manner (low number and severity of intra- and postoperative complications). In addition, the use of this strategy in the front line serves a multiple purpose: i) it provides a psychological benefit to the patient who feels “treated”, ii) it provides symptomatic relief in the presence of symptoms (rather frequent in this subgroup), iii) it provides enough tissue sample that may be used for guidance on the heterogeneity of the disease (usually present) and on the morphological, immunohistochemical and molecular variants involved, that commonly serves as a guide for subsequent systemic treatment (agent and treatment sequence among the many currently available), iv) reduces the burden of disease and thus facilitates the fight of the patient's immune system against the malignancy, thus avoiding the immunological sink generated by the overwhelming burden of disease and enabling an effective immune response, and v) prolongs the survival of the patient, that otherwise left to its free evolution would present a fatal outcome in a short period of time.

We agree with the reviewers´comments. For this reason, we have included in the introduction some notes that summarize the abovementioned precepts and would lead the potential reader in a quick and detailed way towards the objectives pursued by the study. Corrections have been made in the Introduction according to the suggested recommendations (see lines 14-25 in the Introduction section).

Conclusion should highlight the major outcome of study and how it will be contributing in the advancement of researcher area

The main results of the study are: i) the surgical procedure is feasible despite being complex, ii) the techniques derived from transplant surgery can be used in these patients with acceptable morbidity and mortality rates, iii) in our experience, the use of the surgical strategy in the front line relieves the symptoms in the majority of the patients when present, iv) it also provides an optimal tissue sample to be used for pathological and molecular analysis allowing in turn some guidance on the treatment regime/schedule to be used in the necessary subsequent systemic treatment, thus obtaining a more directed, precise and comprehensive treatment approach for the patient aiming to improve the chance for long-term survival, and v) provides modest but lasting responses in survival per se.

We agree with the reviewer's comment. We have added a series of notes to complement what has already been established in the conclusions (see lines 3-8 in the Conclusions section).

Figure 2 and 3 can be re-organized.

We tried to re-organize the figures as per reviwer´s request. However, the re-organized versions obtained prevented adequate visualization of the figures. Therefore, we maintained the figures in its original version.

Optional to author, it will be more easy to understand the goal of the study if author add a scheme or figure presenting overview of the present study at end of introduction.

We agree with the reviewer´s comment. We added some notes to complete the information provided in the Introduction section and to provide the reader with a clearer picture on the purpose of the study in advance (see lines 29-33 of the Introduction Section)

Reviewer 2 Report

The authors have reviewed their experiences in 18 patients of cytoreductive nephrectomy. Although they report high success rates, there are no unique surgical techniques they have provided in this manuscript. Moreover, the success rates in our institution is similar to what they have reported.

Current topics that are interesting would be the selection of patients who will benefit for cytoreductive nephrectomy, or adjuvant systemic therapy after nephrectomy. However, none of these topics have been handled in this study. They have merely reviewed the current understanding in this field, not discussing the novel results they have provided. Thus, the originality of this paper is too low to be considered in this journal. 

Author Response

Reviewer #2

The authors have reviewed their experiences in 18 patients of cytoreductive nephrectomy. Although they report high success rates, there are no unique surgical techniques they have provided in this manuscript. Moreover, the success rates in our institution is similar to what they have reported.

We thank the reviewer for taking the time to review our manuscript. It is indeed a series of 18 patients in whom cytoreductive nephrectomy has been performed. The techniques used in the surgical treatment of these patients are not unique from the point of view of their originality. However, the surgical treatment of this type of patient represents a challenge for the urologist because of its complexity. There is no standardized technique for the treatment of all cases. On the contrary, the techniques used represents a set of tools and maneuvers, some of them of considerable complexity, which allow the patient to be free from the disease when they are used in the proper number and sequence. The technique in our opinion should be tailored to the particular case. Of note, we have been using the techniques derived from transplant surgery applied to complex renal cell carcinoma cases since the late 1990s with great success. Many of the maneuvers used here are part of our original description. However, the aim of the paper is not to propose a new technique for the treatment of the patient, but to report the experience obtained using the techniques we describe for the treatment of a subgroup of extremely complex patients, in whom the lack of experience regarding the results after surgery makes it difficult to make decisions on which or in what sequence the available therapeutic tools should be used.

According to the results obtained in our study, we can conclude that transplantation techniques can be safely used in this type of patients. In addition, surgery aimed at eliminating all vestige of disease may constitute an attractive therapeutic modality to be used in the front line with a triple intention: i) to reduce the tumor burden improving patient's immune response to the disease, ii) to obtain sufficient sample of tissue to properly address the histological subtypes involved (and their genetic and molecular variants) in a disease that is already heterogeneous in order to allow a further choice of systemic treatment among the myriad of those available, and iii) to generate a psychological benefit in the treated patient who would also benefit from a symptomatic relief (frequently present).

If the results obtained in the surgical management of these particular cases at the reviewer's institution are similar to those reported here, we are pleased to congratulate him/her on their results as well. However, we do not know on the group to which he/she refers, and we do not know either if the results of this group have been included or not in the discussion section. If it is the reviewer´s wish, we will be glad to include the reference (if available) for appropriate discussion (if we have not already done so).

Current topics that are interesting would be the selection of patients who will benefit for cytoreductive nephrectomy, or adjuvant systemic therapy after nephrectomy. However, none of these topics have been handled in this study. They have merely reviewed the current understanding in this field, not discussing the novel results they have provided. Thus, the originality of this paper is too low to be considered in this journal. 

The main feature of the patients included in this study is not cytoreductive nephrectomy sensu stricto, but its performance in a series of patients who, in addition to presenting vascular involvement, presented invasion of neighboring organs. As the reviewer may remember, vascular involvement only occurs in 4-10% of patients with renal cell carcinoma, while concomitant involvement of neighboring organs is extremely rare. Therefore, series of patients presenting with these two features who have received surgery as a initial therapy are scarce in the literature.

This series, therefore, presents two points that make it quite original in our opinion; on one hand it gathers a representative number of patients harboring an uncommon entity, on the other hand it presents the results obtained after their surgical treatment, a fact that is also infrequent in the literature either. Current knowledge on the topic discussed is based on series like this one. There are no prospective randomized studies available, and therefore there are no recommendations or guidelines with a high level of evidence for the therapeutic management of these patients. For this reason, we believe it is important to present our experience from which others facing similar cases can benefit. In addition, the discussion is therefore limited by the scarcity of published experiences. In our opinion all the representative available knowledge on the topic have been addressed through the manuscript properly.

Finally, the topics that the reviewer may find interesting are not of our concern, nor we do judge them. They are indeed interesting and we hope that he/she will be able to read about them in other manuscripts (of the thousands published daily) soon.

Reviewer 3 Report

This paper reports a very valuable and unique experience about a radical surgery technique on a series of patients (18) with locally advanced RCC invadind the surrounding organs, with or without IVC thrombus. The novelty of the approach consists in part with the application of liver transplantation techniques to aid in the surgical management of the cases. The authors should be praised for the excellent results in reporting this surgical series of cases, all performed by a single surgeon, with obvious extraordinary surgical skills in managing complex open-surgery multi-organ surgery. 

Some points need to be addressed by the authors, before publication of the manuscript:

1. The success of this strategy seems to be the application of manoeuvres derived from transplantation surgery to traditional kidney radical surgery. What are the suggestions of the authors for achieving this kind of expertise in a more general situation? Is a general surgery training necessary? Is a transplantation experience needed? Since experience in transplantation surgery seems to be necessary, what would the authors recommend to surgeons/urologists who want to excel  in this kind of surgery? Or alternatively, should a multi-disciplinary team approach be advocated? Liver transplantation surgeons, general surgeons, urologists, cardio-thoracic surgeons (in case of level IV thrombi), working all together in a multidisliplinary approach? Could this be a possible solution to make this surgery more accessible to large centers, where  multi-disciplinary teams are available?

2. line 104: 9 patients with pancreatic extension of the disease (81%).Why 9 patients are 81% of the 18 total patients? Is 9 patients 50% of the group? The authors should clarify this point.

3. This surgical group of cases establishes the technical feasibility of this complex open multi-organ surgery, with no reported operative mortality, acceptable complication rates and cases of prolonged survival. This is certainly a great achievement. But is this the only goal of this type of surgery? What about quality of life? Do patients, as in this group, with a 44% rate of abdominal pain, a 17% rate of hematuria, a 5.5% rate of intestinal bleeding, have an acceptable quality of life? Does control of symptoms has a role in the decision making when planning this extensive surgery? What do we know about the quality of life of the patients in the study group, as compared to historical series of untreated cases? Shouldn't we take into account also mitigation of severe symptoms?

4. The majority of patients in the group were females, as reported in the text. Interestingly though, in table 1, only the male patients are reported, even if they are a minority. Is there a specific reason for this gender bias? The table and the text shouldn't report the same percentages? The table and the text should report the percentage of the majority of the patients, who are indeed females.

5. I think it is remarkable that 50% of the patients did not require blood transfusions during surgery or in the immediate post-operative period. I think this is a great result. Where there specific manoeuvres utilised to obtain this result? Intra-operative blood recovery? Other anesthesiologic tips and tricks? This would be a relevant piece of information for the readers.

6. Major complications occurred mostly in patients with pancreatico-duodenal involvement, as predictable. However, patients with pancreatico-duodenal involvement seem to live longer (median survival after resection: 36 months) as compared with patients with liver involvement (24 months median survival after resection). 12 months difference in median survival, as reported by the authors, seems to be quite a relevant difference. Any hypothesis on this difference? Were there more cases with positive margins in the cases with liver resection? More positive lymphnodes? Was the liver resection technically more challenging? This points should be addressed in the discussion.

7. The authors state that the main purpose of the study is to review their personal experience in the removal of renal tumours in conjunction or not with IVC thrombus. However, it should be stated also that the purpose of this paper is to show the feasibility and safety of the radical surgical approach in these cases  and to show, more importantly, that this surgery, when properly performed, is worth the effort. It shows curative intent, in experienced hands, and possibly quality of life improvement, in patients with advanced bulky highly symptomatic disease.

8. This study reflects the experience of a single high-volume surgeon from a referral center. The conclusion should state that management of these rare cases should be safely entrusted either to an experienced surgeon with exceptional skills, or to a multidisciplinary surgical team, but exclusively in a high volume referral center, where complex procedures, such the ones reported, can be successfully performed in experienced hands. 

Author Response

Reviewer #3

This paper reports a very valuable and unique experience about a radical surgery technique on a series of patients (18) with locally advanced RCC invading the surrounding organs, with or without IVC thrombus. The novelty of the approach consists in part with the application of liver transplantation techniques to aid in the surgical management of the cases. The authors should be praised for the excellent results in reporting this surgical series of cases, all performed by a single surgeon, with obvious extraordinary surgical skills in managing complex open-surgery multi-organ surgery. 

We appreciate the time spent by this reviewer in evaluating our manuscript. We are glad that this reviewer finds our experience useful and valuable to other groups dealing with these always complex cases. Likewise, we appreciate the reviewer's opinion on our approach to these cases.

Some points need to be addressed by the authors, before publication of the manuscript:

We will try to answer each of the reviewer's observations in a point-by-point manner as required.

  1. The success of this strategy seems to be the application of manoeuvres derived from transplantation surgery to traditional kidney radical surgery. What are the suggestions of the authors for achieving this kind of expertise in a more general situation? Is a general surgery training necessary? Is a transplantation experience needed? Since experience in transplantation surgery seems to be necessary, what would the authors recommend to surgeons/urologists who want to excel in this kind of surgery? Or alternatively, should a multi-disciplinary team approach be advocated? Liver transplantation surgeons, general surgeons, urologists, cardio-thoracic surgeons (in case of level IV thrombi), working all together in a multidisliplinary approach? Could this be a possible solution to make this surgery more accessible to large centers, where  multi-disciplinary teams are available?

In our opinion, there is not a single way to achieve a successful outcome in the surgical treatment of these patients. The primary objective to be achieved is the benefit of the patient in terms of oncological outcomes and quality of life. Perhaps, the way to achieve this results is not so important. This manuscript presents one manner to deal with this reality, but there may be others equally useful that guarantee comparable results.

In this sense, the transplantation-based techniques applied to these difficult and complex cases are an optimal option. In our experience, its use represents an adequate way to treat these masses safely. We have been using these maneuvers since the late 1990s with proven success (in a very large sample of patients) and many of the maneuvers herein used are the result of our own original description. Although transplantation techniques are undoubtedly derived from the experience acquired from the surgical fields of procurement and transplantation of abdominal organs, they are not exclusive to transplantation nor should they be limited to transplantation exclusively. These techniques have also been used successfully in other areas such as surgery for major trauma (with complementary and mutually enriching experiences for both areas) and by surgeons from different specialties (vascular, general, transplantation, urologists, ...). The techniques are absolutely reproducible and can be learned. It is at this point, where its usefulness lies. It is true that many of the maneuvers are difficult to perform, either because of the anatomical situation and the relevance of the structures involved, or because of the characteristics of the particular surgical field (masses of a certain volume compromising vital structures). There are not two identical cases and the decision-making regarding the precise maneuvers to be used in each case must be tailored on a particular basis.

In our opinion, enough experience in open surgery is essential to deal with these cases. Experience in abdominal organ transplantation is highly desirable but not mandatory. These are not recommended cases for the novice surgeon, but under proper supervision, a resident in training can cope with them at the end of the residency. Obtaining the necessary experience inexorably requires a “sufficient” annual number of cases. Given the rarity in its presentation, this number of cases is only possible in referral centers. For this reason, in our opinion the appropriate place for the treatment of these patients are referral centers. On the other hand, only these centers may present the optimal conditions for the management of these patients in a single institution, since all the required resources are available in one place. In this way, each particular case can be discussed directly within a fully involved multidisciplinary group, deciding a priori the number of specialists required to solve the case.

We have referred to these aspects at the end of the discussion section, where the the limitations of the study are mentioned. We have included additional information in this regard in the Conclusions section. However, if it is the reviewer's wish, we could include an additional new paragraph summarizing these notes in the discussion section in a further revision of the manuscript.

  1. line 104: 9 patients with pancreatic extension of the disease (81%).Why 9 patients are 81% of the 18 total patients? Is 9 patients 50% of the group? The authors should clarify this point.

We agree with the reviewer's comment. Indeed, the referred percentage should be 50% (9/18) and not 81% as it appeared in the initial version of the manuscript. It is a typo, which has been corrected in the revised version.

  1. This surgical group of cases establishes the technical feasibility of this complex open multi-organ surgery, with no reported operative mortality, acceptable complication rates and cases of prolonged survival. This is certainly a great achievement. But is this the only goal of this type of surgery? What about quality of life? Do patients, as in this group, with a 44% rate of abdominal pain, a 17% rate of hematuria, a 5.5% rate of intestinal bleeding, have an acceptable quality of life? Does control of symptoms has a role in the decision making when planning this extensive surgery? What do we know about the quality of life of the patients in the study group, as compared to historical series of untreated cases? Shouldn't we take into account also mitigation of severe symptoms?

We agree with the reviewer's comment. Of course, one of the main issues taken into account to offer a surgical strategy to these patients was the possibility of achieving at least partial symptomatic relief that would improve their quality of life.Therefore, we have included the information regarding the symptomatic relief experienced by the patients included after the intervention in the results and discussion sections where appropriate.

  1. The majority of patients in the group were females, as reported in the text. Interestingly though, in table 1, only the male patients are reported, even if they are a minority. Is there a specific reason for this gender bias? The table and the text shouldn't report the same percentages? The table and the text should report the percentage of the majority of the patients, who are indeed females.

We agree with the reviewer's comment. In the initial version of the manuscript, complementary information was provided in the text (section 3.1 Patient Demographics) and Table 1. Perhaps this information should be complete in both places. The data regarding male and female frequencies have been included as required.

  1. I think it is remarkable that 50% of the patients did not require blood transfusions during surgery or in the immediate post-operative period. I think this is a great result. Where there specific manoeuvres utilised to obtain this result? Intra-operative blood recovery? Other anesthesiologic tips and tricks? This would be a relevant piece of information for the readers.

We agree with the reviewer's comment. Although the blood recovery system was available in all cases, it was not necessary in any of the cases included in this series. Transplantation techniques allow us to conduct this type of procedures under conditions of low blood loss even when vascular involvement is present. Preoperative self-donation programs or hemodilution strategies are not used in our routine practice. We did not include additional information regarding this issue. However, if it is the reviewer´s wish we would include a specific note further.

  1. Major complications occurred mostly in patients with pancreatico-duodenal involvement, as predictable. However, patients with pancreatico-duodenal involvement seem to live longer (median survival after resection: 36 months) as compared with patients with liver involvement (24 months median survival after resection). 12 months difference in median survival, as reported by the authors, seems to be quite a relevant difference. Any hypothesis on this difference? Were there more cases with positive margins in the cases with liver resection? More positive lymph nodes? Was the liver resection technically more challenging? This points should be addressed in the discussion.

We agree with the reviewer´s comment. The differences in terms of survival between these two subgroups seem notable in favor of the subgroup with exclusive pancreatico-duodenal involvement. Although in cases of pancreatico-duodenal involvement, postoperative complications are more frequent, the actuarial survival observed at 5 years is around 15% higher. These results probably reflect differences in the pattern of spreading between these two subgroups. While pancreatic-duodenal involvement occurs mainly by direct contiguity, liver involvement can result from direct invasion or from hematogenous dissemination of the disease. Both patterns can coexist in the same patient. Probably, hematogenous disemination would carry worse prognosis and therefore shorter survival rates. Conversely, in cases in which pancreatic resection affected the cephalic region, surgical reconstruction after removal is technically more demanding and, with no doubt, more subject to complications than resection of a portion of the liver parenchyma. However, intraoperative bleeding in these latter cases is more important and more difficult to manage intraoperatively.

We added a complete paragraph on our posible explanation of this issue as per reviewer´s request (see Discussion section lines 66-80).

  1. The authors state that the main purpose of the study is to review their personal experience in the removal of renal tumours in conjunction or not with IVC thrombus. However, it should be stated also that the purpose of this paper is to show the feasibility and safety of the radical surgical approach in these cases  and to show, more importantly, that this surgery, when properly performed, is worth the effort. It shows curative intent, in experienced hands, and possibly quality of life improvement, in patients with advanced bulky highly symptomatic disease.

We agree with the reviewer's comment. According to the reported results, the procedure is feasible and worth the effort, since it aims a curative intent and also provides a better quality of life through symptomatic relief. We have included a few lines in the revised version of the manuscript reinforcing these ideas as per reviwer´s request (see lines 14-18 of the discussion section).

  1. This study reflects the experience of a single high-volume surgeon from a referral center. The conclusion should state that management of these rare cases should be safely entrusted either to an experienced surgeon with exceptional skills, or to a multidisciplinary surgical team, but exclusively in a high volume referral center, where complex procedures, such the ones reported, can be successfully performed in experienced hands. 

We agree with the reviewer´s comment. In this way, we included some lines in the Conclusions section regarding this issue (see Conclusions section lines 11-15)